# Acidification Effects on In Situ Ammonia Emissions and Cereal Yields Depending on Slurry Type and Application Method

Christian Wagner [1], Tavs Nyord [2], Annette Vibeke Vestergaard [3], Sasha Daniel Hafner [2] and Andreas Siegfried Pacholski [1,4,*]

[1] Institute of Ecology, Leuphana University of Luneburg, Scharnhorststrasse 1, 21335 Luneburg, Germany; cwagnerscience@gmail.com

[2] Department of Biological and Chemical Engineering, Finlandsgade 12, Aarhus University, 8200 Aarhus, Denmark; tavs.nyord@eng.au.dk (T.N.); sasha.hafner@eng.au.dk (S.D.H.)

[3] SEGES, Agro Food Park 15, 8200 Aarhus, Denmark; avv@seges.dk

[4] Thuenen Institute of Climate Smart Agriculture, Bundesallee 65, 38116 Braunschweig, Germany

[*] Correspondence: andreas.pacholski@thuenen.de

**Abstract:** Field application of organic slurries contributes considerably to emissions of ammonia ($NH_3$) which causes sever environmental damage and can result in lower nitrogen (N) fertilizer efficiency. In recent years, field acidification systems have been introduced to reduce such $NH_3$ emissions. However, combined field data on ammonia emissions and N use efficiency of acidified slurries, in particular by practical acidification systems, are scarce. Here, we present for the first time a simultaneous in situ assessment of the effects of acidification of five different organic slurries with a commercial acidifications system combined with different application techniques. The analysis was performed in randomized plot trials in winter wheat and spring barley after two applications to each crop (before tillering and after flag leave emergence) in year 2014 in Denmark. Slurry types included cattle slurry, mink slurry, pig slurry, anaerobic digestate, and the liquid phase of anaerobic digestate. Tested application techniques were trail hose application with and without slurry acidification in winter wheat and slurry injection and incorporation compared to trail hose application with and without acidification in spring barley. Slurries were applied on 9 m × 9 m plots separated by buffer areas of the same dimension. Ammonia emission was determined by a combination of semi-quantitative acid traps scaled by absolute emissions obtained from Draeger Tube Method dynamic chamber measurements. Experimental results were analysed by mixed effects models and HSD post hoc test ($p < 0.05$). Significant and almost quantitative $NH_3$ emission reduction compared to trail hose application was observed in the barley trial by slurry incorporation of acidified slurry (89% reduction) and closed slot injection (96% reduction), while incorporation alone decreased emissions by 60%. In the two applications to winter wheat, compared to trail hose application of non-acidified slurry, acidification reduced $NH_3$ emissions by 61% and 67% in cattle slurry, in anaerobic digestate by 45% and 57% and liquid phase of anaerobic digestate by 58%, respectively. Similar effects but on a lower emission level were observed in mink slurry, while acidification showed almost no effect in pig slurry. Acidifying animal manure with a commercial system was confirmed to consistently reduce $NH_3$ emissions of most slurry types, and emission reductions were similar as from experimental acidification systems. However, failure to reduce ammonia emissions in pig slurry hint to technical limitations of such systems. Winter wheat and spring barley yields were only partly significantly increased by use of ammonia emission mitigation measures, while there were significant positive effects on apparent nitrogen use efficiency (+17–28%). The assessment of the agronomic effects of acidification requires further investigations.

**Keywords:** acidification; slurry; ammonia emission; application method; dynamic chamber; fertiliser; multi-plot field trial; passive sampler; nitrogen use efficiency; yield





## 1. Introduction

Atmospheric nitrogen ($N_2$) is transferred into reactive forms (Nr), e.g., through industrial fixation into ammonia ($NH_3$) (~80 Mt N yr-1). The primary purpose of this nitrogen (N) conversion is to support food production through fertiliser use. About 50% of the current human population depends on synthetic nitrogen fertilisers [1] for food. Nr not taken up by the crop is lost to the environment. The side effects of this Nr in the environment include global warming, acidification of soil, eutrophication of habitats, and water quality deterioration, as well as formation of atmospheric micro-particles [2]. In particular, agricultural $NH_3$ emissions (90% of total European Union $NH_3$ emissions) cause about 45% of airborne eutrophication, 31% of soil acidification, and 12% of fine dust formation within the EU 15 [3]. $NH_3$ emissions also mean a considerable loss of fertiliser nitrogen for the crop [1]. Emissions of $NH_3$ are accountable for the acidification and eutrophication of nitrogen-limited ecosystems [3]. About 40–50% of the global anthropogenic $NH_3$ emissions are related to manure from livestock production [4]. Additionally, less than 50% of the N input in agriculture is utilized in agronomic outputs [5]. A better understanding of the use of organic fertilisers is required to reduce Nr emission. Petersen et al. [6] stressed that farmers have to consider new strategies for manure management to minimise environmental impact and increase fertiliser value.

Ammonia volatilization is influenced by type and composition of slurry, application technique, pH level of soil and manure, as well as environmental conditions, such as temperature, wind and rain [7–9]. Trailing hose application is considered as environmentally friendly application system in many European countries and is applicable in growing crops [10]. However, emissions can still be high, and this calls for further reduction [7]. Further emission reduction can classically be attained by incorporation systems as slurry injection [11] or mixing slurry with surface soil. Arable closed slot injectors use a strong tine to inject slurry at about 0.1 m depth and cover applied slurry with soil immediately after application. Injection has become a common application method in countries with strong environmental regulations on manure use (such as the Netherlands and Denmark) [10], and strong emission reduction was proven in many field trials [7]. In addition, pre-treatment of slurries on-farm can help to reduce ammonia emissions from field applied slurries. Among those are additives (clay minerals, humic acids), solid–liquid separation or manure aeration [12,13]. However, for some of the pretreatments, it is still uncertain whether they can reduce ammonia or other trace gas emissions for the whole slurry life cycle.

As a new approach for emission reduction, slurry acidification was introduced to manipulate slurry pH, which shifts the $NH_4^+/NH_3$ equilibrium and thereby reduces ammonia emissions. All else being equal, a one unit reduction in pH reduces $NH_3$ concentration by a factor of ten [9,10]. In practice, mainly highly concentrated sulphuric acid is used for pH adjustment, thereby adding sulphur as a major plant nutrient. There exist two main approaches: on-farm acidification [14] with higher acid demand and stronger pH stabilization and single point field acidification [15]. The latter adjusts slurry pH to one specific level immediately before application. Commercial suppliers of single point field acidification systems recommend the adjustment of the pH value to 6–6.5 pH to have an $NH_3$ emission reduction of around 50%. The amount of acid needed to reach the target pH value varies depending on slurry properties [16]. Slurries may react differently to acidification due to variable pH buffer capacity [17], initial pH value, and specific composition (e.g., dry matter concentration). Farmers may be motivated to use slurry acidification because of its reduction of $NH_3$ emission while sustaining high manure application efficiency [10], potentially increasing fertiliser N efficiency with lower soil compaction compared to slurry injection [18].

However, mainly laboratory systems and experimental systems with acidification procedures differing from commercially available systems were used to derive ammonia emission reduction factors for acidification and its agronomic effects. While experimental systems adjust slurry pH in the tank some time before it is pumped to the application tubes, acid is dosed instantaneously before slurry release in commercial systems. Slurry pH is

probed within the tube system for acid dosage. It remains an open question whether the performance of such systems is well reproduced by experimental systems.

A straightforward comparison and assessment of acidification systems requires a better understanding of effects of fertiliser type and application technologies under same experimental conditions [19]. Due to methodological limitations, effects of treatments on ammonia emissions are usually assessed for a specific emission source only [6]. Standard quantitative $NH_3$ loss measuring methods require large field areas, expensive equipment, or an electrical supply. Their application in replicated field trials is limited [19] and statistical assessment of agronomic effects are scarce. Generally, four replicates per treatment are chosen in agronomic trials to allow statistical testing, which is usually not achieved in applying standard quantitative methods. Statistical assessments are then made across trials, sites and years. In the present study, the requirement of simultaneous tests of slurries and application systems was considered by quantitative ammonia loss measurement in sufficiently replicated field plots. The same approach was successfully applied by Seidel et al. [15]. However, such a simultaneous assessment of a wide range of slurries and commercial application systems under field conditions was performed for the first time in the present study compared to earlier research.

Manure processing by anaerobic digestion creates new slurry types (anaerobic digestate and its separated liquid fraction), that vary from untreated manure, and it can also bring changes or increases in $NH_3$ and GHG (greenhouse gas) emissions [20]. Manure processing as, e.g., solid-liquid separation, is also increasingly standard in various countries [21]. There exists no published information on slurry acidification effects on ammonia losses and yield performance from such substrates. Due to high pH buffering capacities and initial pH, it is discussed whether acidification is practically feasible applying biogas digestates regarding acid requirements and robustness of the pH reduction. Similarly, there exists no information on how slurry incorporation systems interact with slurry acidification with respect to ammonia emissions and slurry yield effects.

The treatments for this experiment were designed to represent commonly used organic slurries applied to winter wheat and spring barley in Denmark and other northwestern European regions. Five organic liquid fertilisers (dairy cattle, mink, and pig slurry, as well anaerobic digestate and separated anaerobic digestate) were applied evenly by four application methods and their combinations (band spreader, incorporation with rotary cultivator, shallow closed slot injection and single point field acidification) to 9 m × 9 m plots on two locations in four trials. Loss of $NH_3$ was determined under different canopy conditions, i.e., crop heights and leaf areas.

We investigated, for the first time in a single trial design, how slurry acidification in interaction with slurry application techniques and slurry type influence $NH_3$ emissions and slurry N use efficiency. This was performed to give a proper insight in slurry acidification as a $NH_3$ reduction measure and to identify emission mitigation approaches that deliver the highest ammonia emission reduction efficiency. The objectives were to quantify acid addition requirements of the five different slurries for emission reduction and mitigation potentials of combined application methods. Crop yields were determined to assess the plant nutrition effect of reduced ammonia emission by acidification. Guiding hypotheses were:

Ammonia loss reduction effects of the commercial system are similar to experimental systems;
Effectiveness of a fixed acid dose on slurry pH and ammonia emission reductions varies among slurry types;
Ammonia loss reduction significantly translates in higher yields and N use efficiency;
Combining slurry acidification with slurry incorporation systems for $NH_3$ emission reduction yield stronger $NH_3$ loss reductions and thereby more pronounced yield effects.

## 2. Materials and Methods

### 2.1. Experimental Sites

Four slurry application trials were carried out in 2014: two trials at the beginning of April and two at the beginning of May. Within each experiment, three to ten treatments

were tested. The experiments were performed with two applications to spring barley (*Hordeum vulgare* L.) on the first trial site and two applications to winter wheat (*Triticum aestivum* L.) on separate plot areas on the second trial site. Two slurry applications to spring barley were performed on the same yield trial, resulting in altogether 3 trials on yield effects of slurry types and management. Winter wheat was sown in September 2013. The two fields were located close to the Research Centre Foulum, Central Jutland, Denmark. The podzol soils (World reference base) in spring barley and winter wheat had a loamy sand texture with no crop residues (Table 1) and no tillage less than 1 week before slurry application.

**Table 1.** Soil properties.

| Location (Coordinates) | Texture (% 0–0.25 m) | | | | | Bulk Density (g cm$^{-3}$) | pH |
|---|---|---|---|---|---|---|---|
| | Clay | Silt | Fine Sand | Coarse Sand | TOM * | | |
| First trial (56°28′58.1″ N 9°34′26.9″ E, 500 m distance: 56°29′00.2″ N 9°34′57.3″ E) | 7.7 | 9.8 | 47.8 | 29. | 5.8 | 1.32 | 5.6 |
| Second trial (56°28′53.8″ N 9°36′28.4″ E) | 6.9 | 8.7 | 49.2 | 29.4 | 5.5 | 1.30 | 5.7 |

* total organic matter.

Soil textures did not vary significantly within fields due to a flat terrain with equally distributed soil conditions and properties (~5 ha for each experiment). Both study areas were located within a distance of ~1.8 km, and climate is characterised as a temperate coastal maritime climate with mean annual precipitation of ~704 mm [22,23]. Due to cool temperatures and regular rainfall events before application, soil surfaces were moist at slurry spreading.

### 2.2. Experimental Set-Up

Slurry fertilizer effects on crops and ammonia volatilization losses were quantified simultaneously across treatments using a randomized block design (*n* = 4). Experimental plots were 9 × 9 m each, arranged with an interspace of 9 m to each other. (Figure 1).

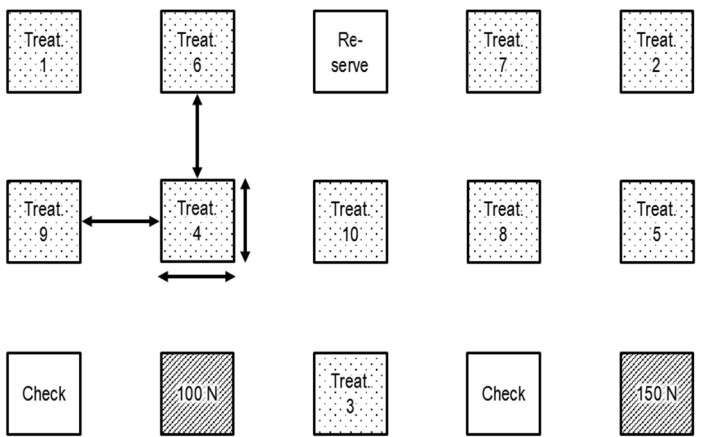

**Figure 1.** Experimental design of one block, used for multi-plot NH$_3$ loss measurements with passive samplers by square treatment plots (9 m × 9 m), Plots were separated by 9 m non-fertilised buffers. Treat. = treatment; Check = control plot with background measurement; 100/150 N = 100/150 kg N ha$^{-1}$ synthetic CAN fertilizer; arrows = 9 m distance.

#### 2.2.1. Slurry Types

On the spring barley field, a single cattle slurry was investigated with five application methods, whereas on the winter wheat field only trail hose application was used, applying five slurries both with and without acidification (Tables 2 and 3). Slurry applications were based on the same total ammoniacal N (TAN) application rates.

**Table 2.** Slurry types, slurry characteristics and application dates, methods and conditions of four slurry applications carried out on spring barley and winter wheat at Foulum Research Centre, Denmark (2014).

| Experiment | Slurry Type | Application Method | Rate | pH | | $H_2SO_4$ | $NH_4^+$-N | DM | DM ac |
|---|---|---|---|---|---|---|---|---|---|
| | | | t ha$^{-1}$ | | ac * | kg m$^{-3}$ | kg ha$^{-1}$ | % | % |
| Application at seeding of spring barley 1.4.14 (soil moisture wet) | CS | Trail hose (th) | 53.0 | | | | 84.8 | 6.5 | |
| | | Incorporation (inc) | 53.0 | 7.2 | | | 84.8 | 6.5 | |
| | | inc + ac | 53.0 | | 5.6 | 2.5 | 84.8 | | 6.4 |
| | | Injection (inj) | 53.0 | | | | 80.0 | 6.4 | |
| Application on standing spring barley 1.5.14 (soil moisture wet) | CS | Th | 53.0 | 7.5 | | | 84.8 | 6.5 | |
| | | th ac | 53.0 | | 5.7 | 3.0 | 84.8 | | 6.5 |
| First winter wheat trial at the end of tillering stage 3.4.14 (soil moisture wet) | CS | th +/− ac | 53.0 | 7.3 | 6.5 | 4.0 | 84.8 | 6.2 | 5.7 |
| | AD | | 53.0 | 7.6 | 6.5 | 8.0 | 79.5 | 3.5 | 4.3 |
| | MS | | 37.5 | 7.2 | 6.0 | 3.4 | 78.8 | 1.4 | 1.5 |
| | PS | | 31.7 | 7.1 | 5.4 | 2.0 | 84.4 | 2.8 | 3.0 |
| | AS | | 53.0 | 8.0 | 6.6 | 7.1 | 79.5 | 2.4 | 2.7 |
| Second winter wheat trial at the beginning of flag leaf emergence 30.4.14 (soil moisture wet) | CS | th +/− ac | 53.0 | 7.4 | 6.2 | 4.0 | 84.8 | 6.2 | 5.7 |
| | AD | | 53.0 | 6.7 | 3.2 | 8.0 | 79.5 | 3.5 | 4.3 |
| | MS | | 37.5 | 7.0 | 6.1 | 3.4 | 78.8 | 1.4 | 1.5 |
| | PS | | 31.7 | 6.8 | 5.6 | 2.0 | 84.4 | 2.8 | 3.0 |
| | AS | | 53.0 | 7.9 | 6.5 | 7.1 | 79.5 | 2.4 | 2.7 |

* Target pH 6.0; ac, acidification; AD, anaerobic digestate; AS, liquid phase separated anaerobic digestate; CS, cattle slurry; MS, mink slurry; PS, pig slurry.

**Table 3.** Slurry composition for the acidification treatment experiment, 2014 (winter wheat site).

| | Application Date | Slurry Type | | | | |
|---|---|---|---|---|---|---|
| | | CS | AD | MS | PS | AS |
| $NH_4^+$ (kg $NH_4^+$-N/m$^3$) | 03.04./30.04. | 1.6 | 1.5 | 2.1 | 2.6 | 1.5 |
| DM * (%) | 03.04./30.04. | 6.2 | 3.5 | 1.4 | 2.8 | 2.4 |
| pH | 03.04. | 7.4 | 6.7 | 7.0 | 6.8 | 7.9 |
| | 30.04. | 7.3 | 7.6 | 7.2 | 7.1 | 8.0 |

* dry matter content; CS, cattle slurry; AD, anaerobic digestate; MS, mink slurry; PS, pig slurry; AS, liquid phase separated anaerobic digestate.

In winter wheat the following slurries were included: dairy cattle slurry (CS), co-fermented anaerobic digestate (AD), liquid phase separated anaerobic digestate (AS), mink (MS) and pig (PS) slurry. Due to higher TAN concentrations compared do CS and digestates, mink and pig slurry were applied with smaller volumes. The latter two slurries were also characterized by low DM and viscosity compared to the other slurries. The slurries were similar to other liquid manures in Denmark and were produced at the Foulum Experimental Centre, sourced from farmers close to the research area and the Foulum biogas plant. The biogas slurry (non- and separated anaerobic digestate) was a co-fermented liquid manure, fermented with cattle solid manure, chopped straw and pasture grass.

### 2.2.2. Slurry Fertilization and Application Methods

Five strategies for CS application (Table 2) were investigated at two application dates in spring barley: before sowing (1 April 2014) and at 4-leafs stage (1 May 2014):

(1) Trail hose application followed by incorporation within 4 h by power harrow (inc), before sowing;
(2) Trail hose application of acidified CS, followed by incorporation within 4 h by power harrow (inc ac) before sowing;
(3) Shallow closed slot injection (inj) before sowing;
(4) Trail hose application (th) at EC 22;
(5) Trail hose application + acidification (ac) at EC 22.

For the $NH_3$ measurements before sowing, 2 plots (9 m $\times$ 9 m) with trail hose applied CS were added for the derivation of the ammonia emission transfer factor [24] and as reference for emission reduction by mitigation techniques. Slurry application rate was the same as in the other CS treatments. These two plots were not included in yield evaluation.

In winter wheat, two different slurry application dates were tested: 3rd and 30th of April. Both trials were located on the same field. The first application was performed at the end of stem elongation stage, while the second application was performed at flag leaf emergence. Altogether 10 slurry treatments (5 slurries $\times$ 2 acidification levels) were tested (Table 2). Sulphuric acid dosage was determined from titration measurements performed prior to field application, but the resulting pH in the field was often different from the target of 6.0.

All slurry treatments also received a second N dosage of Sulphur amended Calcium Ammonium Nitrate (CAN 24-0-0-6), 25 kg N/ha in spring barley and 30 kg N/ha in winter wheat. CAN was applied when slurry was fertilized in the respective other barley or wheat trial. As a result, in all slurry treatments, about 110 kg/ha of mineral N was applied, mainly as ammoniacal nitrogen.

In both, barley and winter wheat, two N levels of Sulphur amended CAN (0, 100 and 150 kg N/ha) were included as a reference and for evaluation of nitrogen use efficiency of tested slurries. Plots with no N fertilisation were included for N control, NUE and $NH_3$ background concentration measurements.

Trail hose application was performed with a commercial system (Samson, SB 16–24, folded at 16 m), reduced to the operating width of 9 m. The closed slot injection machine had a width between the tines of 0.06 m and an injection depth set between 0.10–0.13 m.

In-field slurry acidification was performed with a commercial single point acidification system (Gødningsudstyr, Kyndestoft Maskinfabrik ApS, Vesterled 38 A, 7830 Vinderup, Denmark), mixing slurry with 50% sulphuric acid. The acid was pumped from a tank to the application tubes. The mixture was then directly applied to the soil.

For slurry analysis, samples of all slurries were taken at every application at trail hose openings. These samples were analysed for TAN. (Tables 2 and 3).

The application rates of slurry were based on slurry ammonium N concentrations and standard slurry application rates in Denmark.

### 2.2.3. Ammonia Emissions

For determination of $NH_3$ loss, the plot centres were equipped with passive flux samplers, first applied in the standard comparison method (SCM) [25] but used here in combination with a calibrated dynamic chamber method (Dräger Tube Method (DTM), [26] for scaling of the semi-quantitative loss obtained by the passive flux samplers with a transfer factor derived from the DTM to absolute losses [15,27,28]. Transfer coefficients were obtained by the simultaneous measurement with passive sampler and DTM on 2–4 treatment plots with trail hose application and 2 unfertilized control plots by simultaneous measurements with both methods. The measurements on the unfertilized plots are used for the quantification of $NH_3$ background concentrations and identification of eventual drift of emitted ammonia between treatment plots. Areas dedicated for harvest within each plot were not to be touched by the chamber and passive sampler measurement procedures.

Emission measurements started immediately after application. In each experiment, slurry treatments were applied sequentially, each slurry type after another, with slurry application finalized before afternoon. Plots receiving CAN were not included in the measurements, due to the low emissions potential connected with this fertilizer [29,30]. Emission reduction effects are expressed in relation to emissions from the respective reference treatment, which was cattle slurry applied by trail hoses for the barley trial, and the respective unacidified slurry in the winter wheat trials.

The passive sampler was located in the centre of a square plot at the height of 0.15 m above soil or canopy, directly after slurry application and contained 20 mL of 0.05 M sulphuric acid for $NH_3$ absorption. The solution in the samplers was exchanged every 3 to 4 h and was later analysed for ammonia concentrations with an ammonia-electrode

(Thermo Scientific, Beverly, MA, USA). Samples were directly moved to the lab and kept frozen ($-18$ °C) until measurement. Measurements with DTM were made at least at two locations within a plot per measurement date. In detail, stainless steel rings were placed into the upper soil level of one plot for each treatment immediately after slurry application. Two were placed in the slurry application and two between the slurry strips to gain representative slurry coverage. By experience, band applied slurries cover about 50% of the treated area [28]. Each time four chambers were inserted and connected to an $NH_3$ indicator tube and an automatic pump to ensure a defined flow rate through the dynamic chambers. Raw data obtained from dynamic chambers measurements were corrected to account for ambient wind conditions by a calibration formula [26]. Ammonia emission measurements lasted until no fluxes could be detected, between 72–96 h.

The resulting $NH_3$ fluxes of the calibration plots allowed the calculation of cumulative N-loss for each treatment plot. The transfer coefficient is derived as average from 2 (barley trials)–4 (winter wheat trials) plots. Due to the large set of ammonia data only cumulative losses are presented. Emission dynamics are shown in the Supplementary Material.

### 2.3. Meteorological Measurements

A local weather station was placed in the field throughout the entire trial period. Following data was recorded: air temperature, wind speed, wind direction, and relative humidity. The weather station was fitted at the experimental site, which included a 2D-axis ultrasonic anemometer at the height of 2 m, providing wind speeds (0–60 m s$^{-1}$) and wind direction data (WindSonic4 Gill Ultrasonic Anemometer, model: SDI12 OPT4, Gill Instruments Limited, Hampshire, United Kingdom). Further, a thermo-hygro sensor (CS215 Temperature and RH Probe, Campbell Scientific, Logan, Utah, United States of America) was set at 1 m height for air temperature and air humidity.

Precipitation was not measured in the fields. Additional rainfall data were obtained from Foulum weather station.

### 2.4. Yield and Nitrogen Use Efficiency

Cereal yields were determined by harvest of 13.5 m$^2$ with a plot scale combine harvester (Haldrup C-85, InotecGmbH, DK 9670 Løgstør). Cereal yields are reported on a basis of 14% water content (t ha$^{-1}$). Nitrogen yield is quantified by grain protein concentration (% of DM) and N uptake by grain (kg N ha$^{-1}$). The plant samples were dried at 80 °C for 24 h to provide the dry matter (DM) yield. The total plant N content was analysed from the oven-dried material after burning the material at 900 °C, where the N-oxides and molecular N was determined by LECO TruSpec CN (St. Joseph, Michigan, MI, USA) as described in [31].

N use efficiency is quantified and assessed by the calculation of apparent nitrogen recovery efficiency (ANRE, [32], Equation (1))

$$\text{ANRE (\%)} = (N_F - N_{control})/N_{fert} \times 100 \tag{1}$$

$N_F$ = sum of N uptake by grain and stem (kg N ha$^{-1}$) of fertilized treatment;
$N_{control}$ = sum of N uptake by grain and stem (kg N ha$^{-1}$) of unfertilized control;
$N_{fert}$ = N applied by fertilizer (kg N ha$^{-1}$).

### 2.5. Statistics

Ammonia emission data were analysed by a mixed effects model and HSD post hoc test, separately for each slurry application campaign. Variables included in the models differed between summer barley and winter wheat, generally "block" was treated as random effect, while treatment (combination different factors e.g., slurry type +/− acidification), slurry type, acidification and application date were included as fixed effects. The response variable was plot-level total $NH_3$ emission for the complete trial duration. Yield data were analysed by a stratified statistical approach. For both, winter wheat and summer barley, as the first, step a one-way mixed effects model including all fertilizers and treatments as fixed factor and block as random factor were applied. In case of a significant effect

of a fixed factor, a Tukey post hoc test to differentiate treatment levels was carried out with an alpha <0.05. As the study design was not balanced, data subsets were created to analyse the effects of slurry application date, slurry type, and acidification as fixed factors in a multi-factorial mixed effects model for the winter wheat trials. For this analysis only slurry treatments, i.e., 5 slurries with and without acidification at the two application dates were selected. In the case of summer barley, two subsets were created. First, to test the effect of acidification in interaction with application technique, only treatments with and without acidification were included in a 2-way mixed effects model analysis, which were a trail hose application and harrow incorporation. To check for the effect of slurry application techniques, only different slurry application techniques treatments without acidification were selected for a one-way mixed effects model. In all analyses, "block" was the single random effect in the models. Tukey post hoc test was used to differentiate between treatment levels. Statistical calculations were performed with the program R (version R-3.0.1) with the package "agricolae" (version 1.2-3). For mixed effects model analysis and following post hoc tests, the R-packages "nlme" and "emmeans" were used (R package version 3.1-152).

## 3. Results

### 3.1. NH$_3$ Emissions

Average wind speed during all NH$_3$ measurement campaigns was about 3.3 m s$^{-1}$ in April and 3.0 m s$^{-1}$ in May. Throughout the experiments, an average temperature of 6.7 °C was measured (see also Supplementary Material). No precipitation occurred within the first days of experimental periods, a part of a 1 h rain (1 mm) event at the beginning of the first barley trial (data Foulum weather station). Emissions in both winter wheat and the second barley trial were by on-setting rainy periods on the third day of emission measurements.

### 3.1.1. Summer Barley Trials

In summer barley all abatement treatments significantly reduced NH$_3$ emissions compared to trail hose application (Figure 2, NH$_3$ emission dynamics Figure S1). Incorporation reduced NH$_3$ emissions by 59%, incorporation with acidification by 89%, while closed slot injection yielded the highest reduction effect of 96%. Setting the incorporation treatment as a reference point, emissions were further reduced by additional acidification by 72% and were not significantly different from the closed slot injection emissions. In the second barley trial acidification reduced NH$_3$ loss of applied CS by 74% compared to the non-acidified variant.

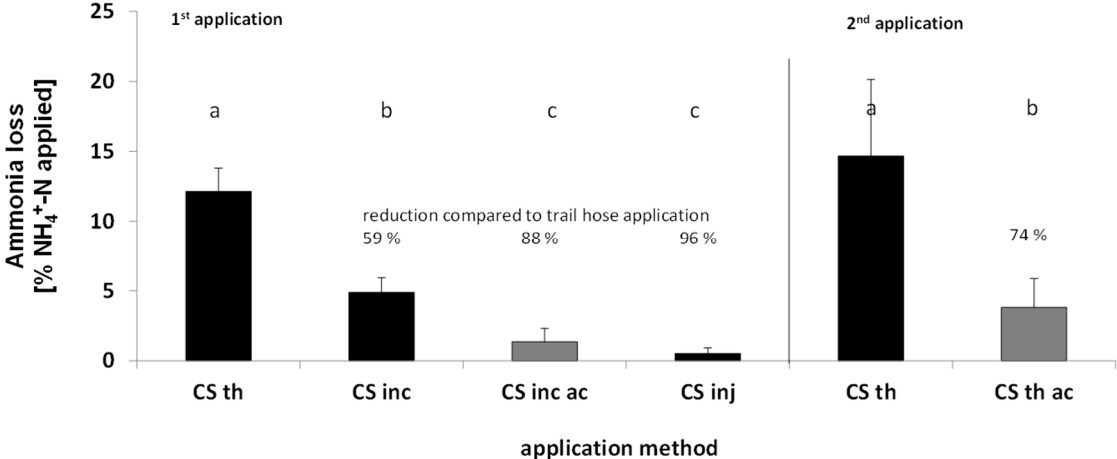

**Figure 2.** Cumulative NH3 losses after application of cattle slurry to spring barley, application rates 84.8 kg TAN ha$^{-1}$ (1st trial), 84.8 kg N ha$^{-1}$. Lowercase letters indicate significant differences between treatments at $p < 0.05$ ($n = 4$ except $n = 2$ for CS th 1st trial; mixed effects model, HSS test $p < 0.05$; lowercase letters indicate significant differences between treatments; CS, cattle slurry; th, trail hoses; inc, harrow incorporation; inj, closed slot injection; ac, acidification).

### 3.1.2. Winter Wheat Trials

Comparing the emissions of the two winter wheat trials, a significantly higher emission level (test not shown) was observed in the first trial, although overall conditions—apart from canopy height—did not differ between trial periods.

In the first winter wheat trial, $NH_3$ emissions from all slurry types were reduced by acidification except for PS (Figure 3, $NH_3$ emission dynamics Figure S2). The multi-factorial statistical test thus resulted in a significant main effect of slurry acidification in interaction with slurry type. Slurry type was also a highly significant factor in the analysis with higher relative ammonia emissions for cattle slurry and anaerobic digestate. Significantly, acidification reduced emissions of CS by 61%, AD by 57%, MS by 74%, and AS by 58%, considering all slurry types individually.

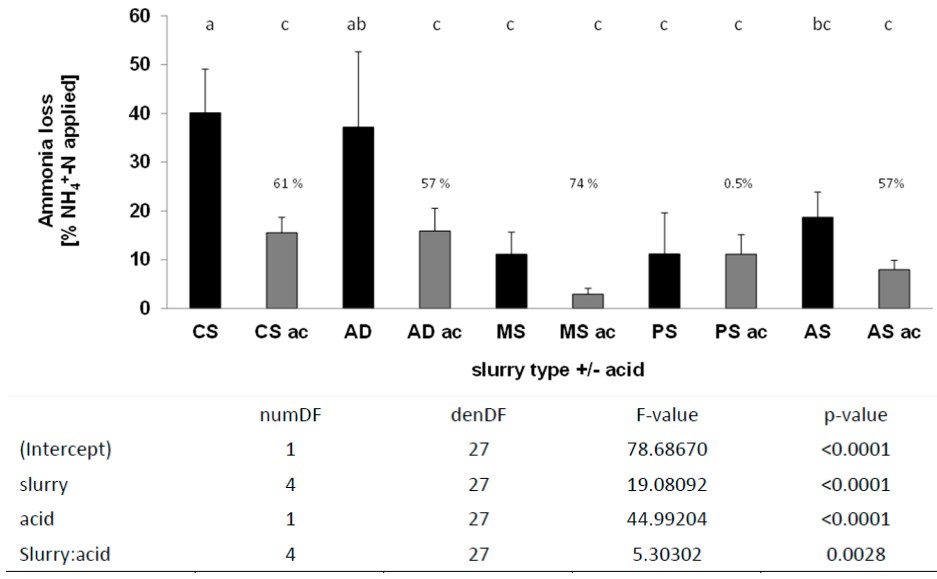

| | numDF | denDF | F-value | p-value |
|---|---|---|---|---|
| (Intercept) | 1 | 27 | 78.68670 | <0.0001 |
| slurry | 4 | 27 | 19.08092 | <0.0001 |
| acid | 1 | 27 | 44.99204 | <0.0001 |
| Slurry:acid | 4 | 27 | 5.30302 | 0.0028 |

**Figure 3.** $NH_3$ loss after application of five different slurries with and without acidification by trail hoses, first winter wheat trial. Applied $NH_4^+$-N kg ha$^{-1}$: CS = 84.8, AD = 79.5, MS = 78.75, PS = 84.42, and AS = 79.5, *n* = 4; mixed effects model, HSD test *p* < 0.05, *n* = 4; lowercase letters indicate significant differences between treatments; CS, cattle slurry; A, anaerobic digestate; MS, mink slurry; PS, pig slurry; AS, liquid phase separated A; ac, acidification.

In the second winter wheat trial (Figure 4), the same main effects were observed in the multivariate model. Losses from MS and PS, as well as AS, were not significantly reduced by acidification when considering separate paired comparisons. However, across all tested slurries there existed a significant main effect of acidification and slurry type. However, there was also a significant interaction between acid and slurry type.

### 3.2. Acid Requirements

On average, separated anaerobic digestate had the highest acid requirement of 4.9 kg $H_2SO_4$ per pH unit per t slurry, followed by cattle slurry (4 kg), mink slurry (3.5 kg), anaerobic digestate (3.3 kg) and pig slurry (2.0 kg), the latter with a rather low acid requirement (Figure 5). Although the target pH value was pH of 6, measured pH values at the trail hose outlet differed considerably from the target value. However, emission reduction values were neither correlated with measured pH values after acidification nor with pH change between acidified and un-acidified slurry or slurry dry matter concentrations. When omitting separated digestate slurries with specifically high acid demand from the analysis, $NH_3$ emission reduction effects of acidification on the remaining four slurries were closely ($r^2$ = 0.98) related to sulphuric acid addition (including separated digestate $r^2$ = 0.63).

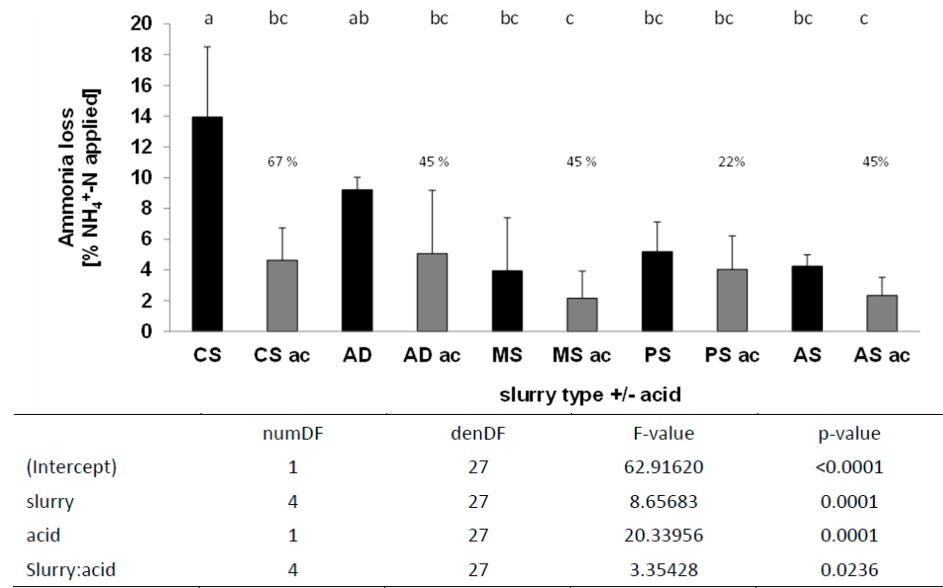

| | numDF | denDF | F-value | p-value |
|---|---|---|---|---|
| (Intercept) | 1 | 27 | 62.91620 | <0.0001 |
| slurry | 4 | 27 | 8.65683 | 0.0001 |
| acid | 1 | 27 | 20.33956 | 0.0001 |
| Slurry:acid | 4 | 27 | 3.35428 | 0.0236 |

**Figure 4.** $NH_3$ loss after application of five different slurries with and without acidification by trail hoses, second winter wheat trial. Applied $NH_4^+$-N ha$^{-1}$ CS = 84.8 kg, AD = 79.5 kg, MS = 78.75 kg, PS = 84.42 kg, and AS = 79.5 kg, $n = 4$; mixed effects model, HSD test $p < 0.05$, $n = 4$; lowercase letters indicate significant differences between treatments; CS, cattle slurry; A, anaerobic digestate; MS, mink slurry; PS, pig slurry; AS, liquid phase separated A; ac, acidification.

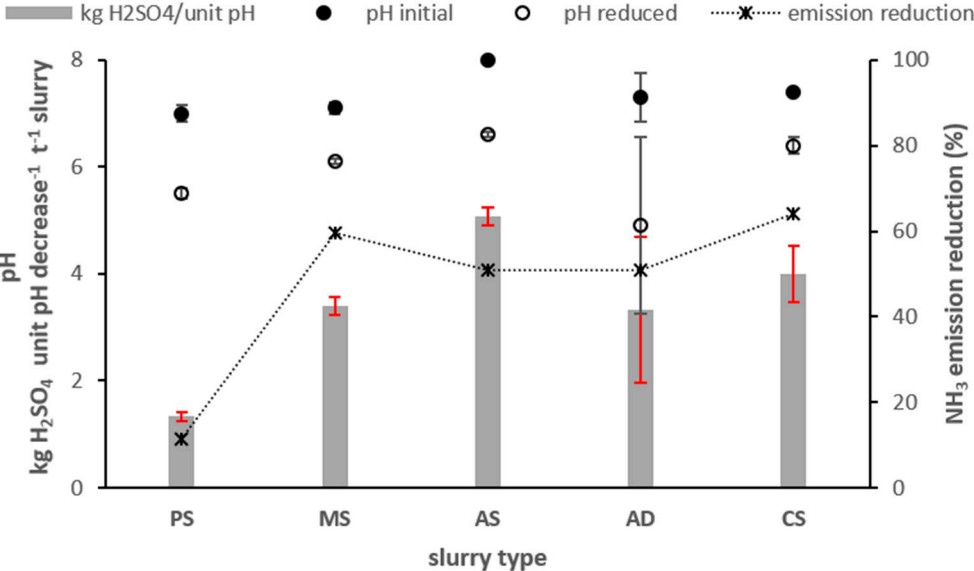

**Figure 5.** pH values before and after single point acidification and acid requirements averaged across both winter wheat trials 2014. Sulphuric acid (50%) was used for pH level reduction, target pH level of 6. PS, pig slurry; MS, mink slurry; AS, liquid phase separated anaerobic digestate; AD, anaerobic digestate; CS, cattle slurry.

### 3.3. Yield Variables

In both crops, yields (Figures were low compared to average grain yield levels in the region, in particular for summer barley. N application levels were chosen based on these yield expectations which probably resulted in excess fertilization. With respect to protein concentrations, summer barley concentrations were high (>10%), indicating low economic value for use of brewing barley, while protein concentration in wheat were low (~10%), indicating poor economic return due to poor grain quality.

Neither in barley nor in the two winter wheat trials did single fertilizer treatments show significant effects on yield compared to the other fertilizer treatments, with exception of incorporation of acidified slurry by harrow in summer barley (Figures 6 and 7). In addition, the CAN treatments resulted in no significantly higher yields. However, when analysing the data by multi-factorial models, in particular, acidification showed significant main factor effects on yield variables (Tables 4 and 5, all data in Supplementary Table S1).

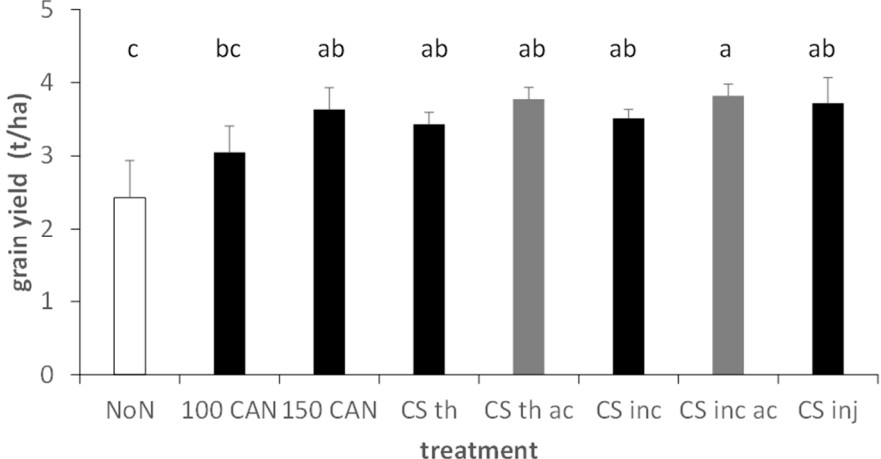

**Figure 6.** Grain yield in the different treatments in summer barley, Foulum, DK, grey colours indicate acidified variants. Fertilization level of applied slurries: 110 kg mineral N. Error bars depict standard deviations. Mixed effects model analysis with HSD post hoc test at *p* < 0.05. Lowercase letters indicate significant differences between treatments; ac, acidification; CS, cattle slurry; th, trail hose; inj, closed slot injection; inc, incorporation with harrow; 100/150, kg N/ha applied as CAN.

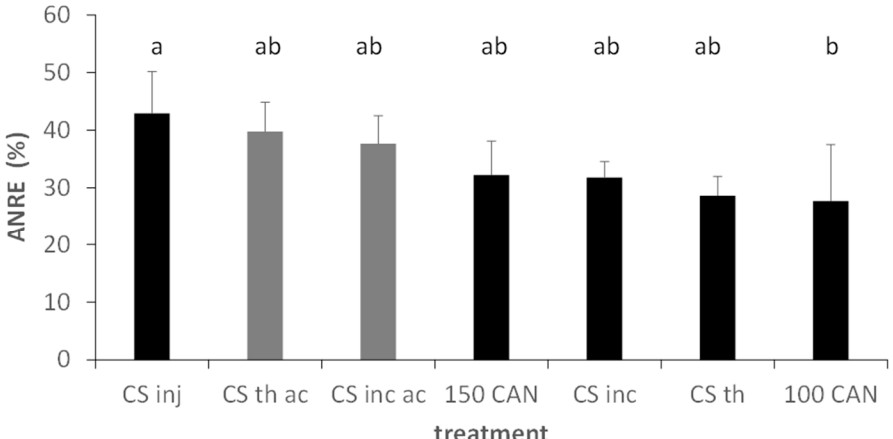

**Figure 7.** Apparent Nitrogen Recovery Efficiency (ANRE) in the different treatments in summer barley, Foulum, DK, grey colours indicate acidified variants. Fertilization level of applied slurries: 110 kg mineral N. Error bars depict standard deviations. Mixed effects model analysis with HSD post hoc test at *p* < 0.05. Lowercase letters indicate significant differences between treatments; ac, acidification; CS, cattle slurry; th, trail hose; inj, closed slot injection; inc, incorporation with harrow; 100/150, kg N ha$^{-1}$ applied as CAN.

**Table 4.** Yield effects of cattle slurry spread with 3 different application methods with and without acidification (sulphuric acid) to summer barley. * significant main effect. alpha < 0.05, ns, nonsignificant. Mixed effects models with HSD post hoc test (*p* < 0.05), different letters indicate significance levels (values level "a" > values level "b").

| | Model | | | |
|---|---|---|---|---|
| | **y~tec × acid, Random = ~1 ǀ Block** | | **y~tec, Random = ~1 ǀ Block** | |
| **Variable** | **Main Effects** | **Factors §<br>(Post hoc Test)** | **Main Effects** | **Factors §<br>(Post hoc Test)** |
| Grain yield<br>(t/ha) | tec ns<br>acid * | ac+: a (+28%)<br>ac−: b | tec: ns | - |
| Ngrain<br>(kg N/ha) | tec ns<br>acid * | ac+: a (+9%)<br>ac−: b | tec: ns | - |
| ANRE<br>(%) | tec: ns<br>acid: * | ac+: a (+9%)<br>acid−: b | tec: * | th: b<br>inc: ab<br>inj: a |
| Ntot<br>(kg N/ha) | tec ns<br>acid * | ac+: a (+10%)<br>ac−: b | tec: * | th: b<br>inc: ab<br>inj: a |
| Protein<br>(%) | tec: ns<br>acid: ns | | tec: * | th: b<br>inc: ab<br>inj: a |
| Data source | Slurry treatments with + and − acid<br>(injection excluded) | | Application methods without<br>acid treatment | |

§ acid = acidification (ac+, with acid; ac−, without acid); tec = application technique (th, trail hose; inc, incorporation by harrow; inj, closed slot injection).

**Table 5.** Yield effects of 5 slurry types applied with and without acidification (sulphuric acid) to winter wheat at two different application dates. Mixed effects models with HSD post hoc test (*p* < 0.05), * = significant main and interaction effects, ":" between factors indicate interaction effects, different letters indicate significance levels (values level "a" > values level "b").

| | Model: lme (y~fert × acid × date, Random = ~1 ǀ Block) | |
|---|---|---|
| **Variable** | **Main Effects** | **Factors § (Post hoc Test)** |
| Grain yield<br>(t/ha) | fert: ns<br>acid: ns<br>date: * | date: early: b, late: a |
| Ngrain<br>(kg N/ha) | fert: *<br>acid: *<br>date: *<br>fert:acid: *<br>date:fert: * | fert: M: a, P: ab, AS: abc, C: bc, A:c<br>acid: +:a (+9%), −:b<br>date: early: b, late: a |
| ANRE<br>(%) | fert: *<br>acid: *<br>date: ns | fert: M: a, P: a, AS: ab, C: ab, A:b<br>acid: +:a (+17%), −:b |
| Ntot<br>(kg N/ha) | fert: *<br>acid: *<br>date: ns | fert: M: a, P: a, AS: ab, C: ab, A:b<br>acid: +:a (+9%), −:b |
| Protein<br>(%) | fert: *<br>acid: *<br>date: * | fert: P: a, M: ab, C: ab, A: b, AS:b<br>acid: +:a (7%), −:b<br>date: early: b, late: a |
| | data: slurry treatments | |

§: *, significant factor; ns, nonsignificant; acid, acidification (+/−); fert, slurry type (M, mink; P, pic; AS, liquid phase anaerobic digestate; A, anaerobic digestate; C, cattle; date, application date (early, late).

### 3.3.1. Barley Trials

In the summer barley trial, acidification resulted in a significant main factor effect with higher values in grain yield (+28%), N grain uptake (+9%), total N uptake (+10%) and ANRE (+9%). Only protein concentration was not significantly influenced (Table 4). The multifactorial analysis of the three application methods in summer barley yielded significantly higher total N uptake, protein concentration and ANRE, while grain yield and grain N uptake remained unaffected. In particular, injection of slurry was more efficient than trail hose application, while there was only a trend for higher values for slurry incorporation with harrow.

### 3.3.2. Winter Wheat Trials

In the winter wheat trial, timing of slurry and CAN application had a significant effect on yield levels (Table 5) with higher yields with the second slurry application. After multivariate analysis of the effects of slurry types and acidification, both showed significant main factor effects on all yield variables excluded grain yield (Figure 8). Acidification resulted in +9% grain N uptake, +9% total N uptake, +7%, protein and +27% ANRE (Table 5, Figure 9). However, there existed interaction effects on grain N uptake between slurry type and acidification and between slurry type and application date. Acidification showed pronounced effects in PS, CS and AD, while only minor effects were observed for mink and, although not significant, negative effect in AS.

When considering ANRE as the main N efficiency measure applied in this study, acidification showed a consistent significant positive effect across both crops and in both wheat trials (Figure 9, Tables 4 and 5).

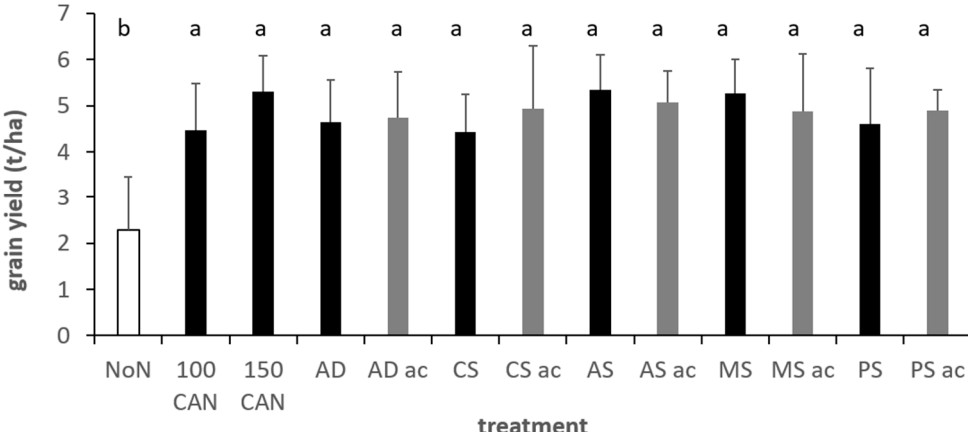

**Figure 8.** Grain yield in the different treatments across two winter wheat trials, Foulum, DK, grey colours indicate acidified variants. Fertilization level of applied slurries: 130 kg mineral N. Error bars depict standard deviations. Mixed effects model analysis with HSD post hoc test at $p < 0.05$. Lowercase letters indicate significant differences between treatments; ac, acidification; MS, mink slurry; PS, Pig; A, anaerobic digestate; AS, liquid phase separated A; CS, cattle slurry; 100/130, kg N ha$^{-1}$ applied as CAN.

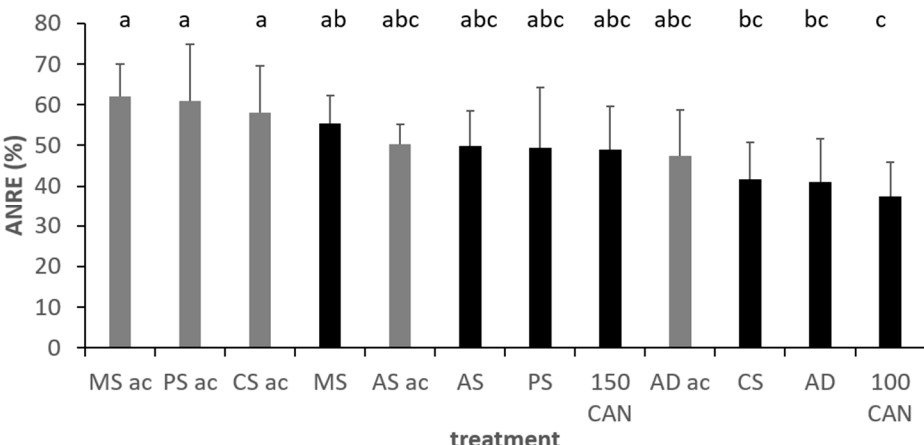

**Figure 9.** Apparent Nitrogen Recovery (ANRE) in the different treatments across the two winter wheat trials, Foulum, DK, grey colours indicate acidified variants. Fertilization level of applied slurries: 130 kg mineral N. Error bars depict standard deviations. Mixed effects model analysis with HSD post hoc test at $p < 0.05$. Lowercase letters indicate significant differences between treatments; ac, acidification; MS, mink slurry; PS, pig; A, anaerobic digestate; AS, liquid phase separated A; CS, cattle slurry; 100/150, kg N ha$^{-1}$ applied as CAN.

## 4. Discussion

### 4.1. NH$_3$ Emissions

The study compared the efficacy of different technological measures to reduce ammonia emissions from field application of several slurry types. The application systems used could all be used with the same operational quality because the sandy soils allowed easy handling of the systems. In contrast to many other slurry acidification trials, a commercial field acidification system was used. Thus, trial performance was technically much closer to farmer's practice and trial results more closely linked to effects potentially observed under practical farming compared to other emission studies.

Ammonia emission measurements had limitations. In between experimental campaigns, generally, transfer factors may change due to effects of environmental conditions on NH$_3$ uptake by the samplers. It therefore has to be determined for each campaign as given in the original study [25]. However, with respect to the derivation of the transfer factor, replicate factors obtained from single plots were not always consistent within an experimental campaign. This can be explained by the uncertainty and variability of fertilizer application at the small scale (<0.1 m$^2$) for DTM dynamic chamber method and within the larger areas used for passive sampling (81 m$^2$). In some cases, the small sampling area was eventually not representative for the whole plot. Therefore, replicate plots were used to determine average transfer factors. Nevertheless, this variation is a source of uncertainty and the number of replicate measurements for derivation of the transfer coefficient was limited. It requires further investigation to which number of replicate microscale measurements are needed to obtain a robust transfer factor.

Overall levels of cumulative ammonia emissions from cattle slurry were within the generally observed range [7]. The measurement approach has been quantitatively validated compared to micrometeorological methods in earlier studies [26–28]. Nevertheless, being a non-standard measurement approach, observed cumulative ammonia emissions from trail hose applied CS of the four emission trials were cross checked with results calculated with the ALFAM2 model [33]. In three of the four trials, measured emissions were close to emissions calculated with the model. However, emissions in the first winter wheat trials measurements deviated strongly from model calculations. These high values were due to high ammonia concentrations in the measurement system and not due to the calibration approach for post-processing for ammonia loss raw data. Nevertheless, emission data are indirectly in agreement with observed yield level differences between the first and the second wheat trials, with higher yield levels in the later application characterized by lower

ammonia emissions. ALFAM2 model calculations can also deviate strongly from emission data measured with standard, i.e., micro-meteorological methods [33].

In the barley trial, cumulative values of the $NH_3$ emissions (Figure 2) of incorporated cattle slurry were significantly higher than the emissions of the same treatment with acidified slurry and closed slot injection. This was probably due to the typical 2 h delay of incorporation after trail hose application. Emissions would have been on the same level, if the slurry was incorporated immediately after application. Following Sommer et al. [10], $NH_3$ emissions would be efficiently decreased by 80% with incorporation directly after surface application.

Injectors significantly lower threat of ammonia loss, with reduction efficiencies of 49–97% compared with 41–48% for band spreading, both relative to surface application [34]. This was also confirmed in our study with simultaneous measurements. Closed slot injectors have drawbacks, the working width is 2–8 m, and the injection is much more energy demanding. For growing crops, the damage from the wheels is higher than when using trailing hoses, and the cutting of roots by the open-slot injection discs can also cause crop damage. Therefore, deep injection is never, and open-slot injection is hardly ever, used in winter cereals. For bare soil, closed-slot injection is often applied while open-slot injection in typical for grassland applications [10,11]. The results showed that emissions from acidified incorporated slurry and closed slot injection were low and not significantly different (Figure 2). However, the cost of acidification and incorporation can be much cheaper for the farmer, than deep injection. Therefore, acidification could play an essential role in $NH_3$ abatement when slurry is applied to bare soil. When surface application is performed in growing cereal crops, open slot injection is costly, due to reasons as earlier described. Field acidification with trail hoses, can be one of the technological alternatives to reduce $NH_3$ loss. Although, acidification increases costs compared to simple trail hose application, due to the requirement for additional equipment and specialized staff handling concentrated acid [10,16]. Conversely, due to cost-effective sulphur (S) fertilization, ease of application, and possible positive yield effects, acidification can be a practical and cost-efficient alternative to injection. Moreover, the reduction effect can be increased by subsequent incorporation, as demonstrated in our study.

The $NH_3$ emissions after trail hose application, incorporation, shallow closed slot injection and single point field acidification varied. Application of slurry with band spreading on the soil beneath a crop canopy can decrease $NH_3$ volatilisation by ~50% compared to surface broadcast application, the efficiency of this technique increases with greater leaf area and height of crop [10,35]. Soil water content, solar radiation, crop height and leaf area significantly influence the potential reduction efficiency of the trail hose technique, with the most significant effect at low soil water content, high solar radiation and great leaf area. [10]. Nevertheless, trail hose application in the second trial (Figure 2), with higher spring barley canopy and a loss of 14% ammonium N applied, was higher compared to trail hose application in the first trial (12%). The soil water content, due to some rain, appears to have affected the results of the first trial of the barley field. As mentioned, rain (1 mm) was observed at the beginning of the first trial. $NH_3$ emissions are at their highest directly after application. Thus, a potential canopy effect was masked by other environmental factors in this trial. For future tests of canopy effects on emissions, doing measurements in manipulated canopies might present better insights.

In general, in both winter wheat trials acidification of slurries showed similar reduction effects on $NH_3$ emissions. However, only small or no reduction effect were observed for PS. Furthermore, acidifying dairy cattle slurry in the barley trial and the early and late winter wheat trials showed a similar reduction effect of 61%, 67% and 74% reduction of $NH_3$ emission, respectively.

Emissions of CS and AD were significantly higher compared to those from other slurries and acidification resulted in significant reduction of losses for both slurries. MS, PS and AS showed no significant differences between each other. This is probably due the low DM concentration compared to the former slurries which were on a similar level. Higher

TAN concentrations may also have played a role (Table 2), as lower volumes of slurry had to be applied and the fraction of infiltrated slurry is higher when the same amount of ammonium N is applied with a smaller volume of slurry. While acidification reduced emissions by about 50% for AS and MS, the effect was almost negligible for PS.

It had to be expected that acidified pig slurry showed stronger reduction effects of about 67% of $NH_3$ emission at a pH of 5.5 [14]. However, in the cited study, on-farm acidification systems were tested. Differences between acidified and unacidified treatments were smallest in PS and emissions were low compared to CS and AD. PS was characterized by the lowest initial pH, low DM concentration and lowest acid demand to reach target pH. Fast infiltration of both acidified and unacidified PS may have dominated the emission process compared to acidification. However, DM concentrations were even low in MS and AS connected with high acidification efficiencies. Hypothetically, the small acid demand entailing eventual fast pH buffer reactions in PS after application may have influenced the effectiveness of acidification. This is supported by the close relationship between acid dosage and reduction effects for the non-digestate slurries. Although, generally, single point acidification seemed to have worked well in four of the five tested slurries, this result may hint to limitations of this acidification approach. On this background, our first hypothesis on the consistent effectiveness of commercial systems across slurry types has to be rejected. More detailed investigation of pH buffer reactions of different slurries may shed more light on this open question. In this context, the high acid demand for pH adjustment for digestate slurries may pose further limitations for the acidification approach due to the high risk of sulphur oversupply.

Regarding sulphur fertilization, crops on to which animal manure is applied often show a lack of S supply because of low concentrations of readily plant-available S. Sulphate in slurry, when stored, is potentially reduced during storage and lost as hydrogen sulphide ($H_2S$). This loss could be reduced by farm acidification [36], and most of the slurry would be still plant available at the time of application [37]. Nevertheless, high demand of sulphuric acid addition, e.g., for anaerobic digestate slurries in this study, may entail the risk of sulphur overfertilization [15] and subsequent sulphate leaching losses. Thus, slurry acidification by sulphuric acid cannot be considered a general solution for reducing ammonia emissions from field applied slurries—keeping sulphur fertilization balances in mind. While injection and taking advantage of cold application conditions in early spring could be a viable mitigation option for small canopies, slurry acidification could be recommended for slurry application later in the vegetation period and in taller canopies.

For all mitigation measures for specific trace gases, there exists the risk that emissions of other trace gases may be concomitantly increased ("emission swapping"). For example, special $NH_3$ mitigation measures during storage of manure can increase $N_2O$ emissions [38,39]. However, for single point acidification no general negative effect on $N_2O$ [40] and $CH_4$ emissions were observed [41] in contrast to higher $N_2O$ emissions found in many cases for slurry injection [40].

However, as part of the risk of excess S fertilization, slurry acidification can mobilize other nutrients and ions which could also have detrimental effects. The beneficial effect of acidification on crop P nutrition is well known [42], but this could potentially also mobilize P in water leaving the root zone, or nickel and zinc, which are used as feed additives transferred to the slurries. These potentially negative side effects of slurry acidification need more attention in research.

The fourth hypothesis is supported by the trial results, as combined mitigation measures resulted in improved $NH_3$ loss abatement in the barley trial and emission reduction in low DM slurries MS and AS in winter wheat. However, considering the PS treatment, the low $NH_3$ emission level of untreated slurry (probably due to low DM) was not further decreased by acidification. The cause for this was probably rooted in unsustained pH decrease, i.e., inappropriate acidification treatment by the used application system.

### 4.2. Acid Requirements

In contrast to on-farm acidification, single point slurry acidification lowers the pH to a target pH value with subsequent pH increase by buffer processes within the slurry [16]. This may entail the risk of an insufficient acid dosage when pH reaction and pH buffer systems are fast as occurred in pig slurry in this trial. Soil texture influence on the result was not covered by this study as all trials were performed on similar soils in the same agro-region. Buffering capacities of slurries are variable, and a variable amount of acid was needed to reduce pH by a certain level (Table 2 and Figure 5), in order to reach the target pH 6 value. The results show that the target 6 pH value was achieved only for MS, while other acidified slurries differed from the target value. A further detailed analysis of slurry properties, with their buffer potential, may give further explanations for the variable acid requirements. DM concentration (Table 2) and slurry processing by biogas production may play a major role. For future trials, it would be advantageous to know more precisely how to adjust the pH value for a particular slurry type, to assure that the acidification system is operating correctly. Therefore, the pH treatment of commercial acidification system needs more attention with respect to in how far this approach is robust enough to guarantee a specific pH level of slurries after leaving the system to the field surface.

The second hypothesis is supported by the results, as the different slurries were characterized by different acid requirements for pH decrease. However, a specific single point pH value at application did not guarantee a specific ammonia emission reduction, as was demonstrated by the pig slurry acidification results in this study. The closest relationship between emission reduction and acidification treatment was to the amount of acid added. Thus, for single point acidification amount of acid addition seems to be eventually more appropriate control measure than target pH value of acidified slurry.

### 4.3. Yield Variables

Emission reduction did not translate in higher grain yields but into significantly higher values for most of the four other tested yield variables. This can be partly explained by the high fertilization level of the trials compared to the realized yields. However, in the treatments with the strongest quantitative reduction of emissions (expressed as kg N per ha emission reduction), CS and AD in the winter wheat and injected CS in the summer barley trial, the increase in yield variables was more pronounced than for the other tested slurries. This may hint to a direct relationship between ammonia emission reduction and increased N uptake by the crop which was also observed for wheat and synthetic fertilizers [19]. Contrastingly, a stronger effect was also observed in PS. In addition to higher N availability after acidification, which did not exist in PS treatment, higher S and P availability [42] can promote crop growth and thereby indirectly cop N uptake. Across all trials, acidification had the most robust and pronounced effects of all application techniques. The specific effect of acidification on yield and N uptake can hardly be generalized as it depends on crop type, fertilization level and soil nutrient supply. Mixed effects were also observed in practical trials around the Baltic Acidification Project [43], but higher winter wheat yields after acidification were particularly observed in Northern Germany close to the sites of this study. Generally, clear NUE effects were observed in Danish trials [18]. The third hypothesis can thus only partly be confirmed, as yield was mainly unaffected by ammonia mitigation measures for single treatments. Nevertheless, a generally positive main effect by acidification on grain yield was confirmed for summer barley and was also found in earlier studies [18].

## 5. Conclusions

The experimental approach used in this work (small replicated plots with a large number of experimental treatments) provided a means for simultaneous comparison of ammonia emissions depending on multiple slurry application technologies under identical conditions. Such a broad approach of simultaneous comparison of application systems and slurry types was performed for the first time in this study. In general, results confirmed

that in-field slurry acidification, slurry injection, and rotary incorporation all significantly reduced $NH_3$ emissions from the field-applied slurry. Furthermore, incorporation plus acidification showed nearly the same reduction in emission as closed slot injection, supporting the hypothesis that a combination of mitigation measures can further reduce $NH_3$ emission. Acidification may be a cost-effective approach for reducing $NH_3$ emission and an alternative to injection depending on slurry type. The results hint to limitations of commercial single point acidification techniques with slurry pH as management variable. Alternatively, baseline acid addition levels seem to be needed to obtain robust emission reduction. Further trials may give further insight into the acid requirements for a significant reduction of $NH_3$ emissions combined, taking into consideration reasonable sulphur fertilisation levels. Results give clear evidence that particular consideration has to be taken concerning acid requirements of different slurry types. From a fertiliser management perspective, it is necessary to use the appropriate amount of organic fertiliser at the right time, to minimise emissions and to be sufficient for crop growth. As in other studies, the yield effect was more pronounced with respect to N uptake and NUE expressed as ANRE. There still exists a gap of published data on yield and NUE effects by slurry acidification in field studies, which call for further investigations.

**Supplementary Materials:** The following are available online at https://www.mdpi.com/article/10.3390/agriculture11111053/s1, Figure S1: time courses of ammonia emissions in the 2 summer barley trials, Folum, DK, 2014; ac acidified, inc incorporated by harrow, inj closed slot einjection, th trail hose, Figure S2: time courses of ammonia emissions in the 2 winter wheat trials, Folum, DK, 2014; ac acidified, Table S1: Grain yield, protein concentrations, nitrogen uptake and apparent nitrogen recovery efficiency in three slurry fertilization trials, Foulum, DK, 2014, $n = 4$, values in brackets indicate standard deviations, letters indicate significance levels after mixed effects model analysis and HSD post-hoc test, $p < 0.05$. AD anaerobic digestate, AS, separated AD, PS pig slurry, MS mink slurry, CS cattle slurry; ac acidified (target pH 6.0) th trail hose, inc incorporation by harrow, inj closded slot injection.

**Author Contributions:** Conceptualization, T.N., A.V.V. and A.S.P.; methodology, A.S.P., C.W. and T.N.; formal analysis, C.W. and A.S.P.; investigation, C.W., S.D.H. and A.S.P.; data curation, C.W. and A.S.P.; writing—original draft preparation, C.W. and A.S.P.; writing—review and editing, A.S.P., C.W., S.D.H. and A.V.V.; visualization, C.W. and A.S.P.; project administration, T.N. and S.D.H.; funding acquisition, T.N. All authors have read and agreed to the published version of the manuscript.

**Funding:** This study was funded by Grønt udviklingsog demonstration program (Gylle-IT), Ministeriet for Fødevarer, Landbrug og Fiskeri-NaturErhvervstyrelsen.

**Data Availability Statement:** A large part of data is available via https://webtrial.dlbr.dk/en/TrialView?country=DK (accessed on 2 August 2021).

**Acknowledgments:** We want to express our thanks to Sven G. Sommer as well as to the supportive team during field trials, in particular Heidi Groenbaek.

**Conflicts of Interest:** The authors declare no conflict of interest.

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
