# Peer review of "Acidification Effects on In Situ Ammonia Emissions and Cereal Yields Depending on Slurry Type and Application Method"

_agriculture, doi:10.3390/agriculture11111053_

Round 1
Reviewer 1 Report
Line 28-29. The reductions presented in this sentences were achieved by trail hose application with and without slurry acidification, separately? The treatments, including NH3 reduction measures and crop seasons were not clearly explained in the abstract.
Line 87-89. But if any negative impacts could be induced by acidification for slurry and soils?
Line 96 and 101. Did you mean that previous field experiments measuring NH3 emissions including limited replicates? How many replicated plots are required for each treatment?
Line 212-218. If these are commonly used by local famers in your study regions?
Lin 256. How about the measurement frequencies? How many days were needed before NH3 emissions from the slurry-added treatments decreased to the background levels?
Line 326. The method of trail hose application is the conventional treatment?
Line 370. Why did you set pH=6 as the target?
Line 384. What were the possible reasons?
Results shown in Tables 4 and 5 are complicated and difficult for the readers to understand.
Line 456. In this section, I think the most important aspects are to discuss the underlying reasons for the reductions of NH3 emissions by application methods and acidification, and also the differences among slurry types. For example, how did the application methods affect NH3 emissions via changing soil NH4+ supply, and how about the soil pH in the treatments with vs. without acidification.
In line 615-618, you indicated that PS exhibited low NH3 emissions and then acidification had no significant impacts. But the readers would more like to learn why PS emitted lower NH3. The effects of slurry type were one of the main research objective of this study. Suggest strengthening discussions on the differences of the internal properties of the slurry.
Line 628. The pH was measured before application, if the target pH had not achieved, why not add more acid?
Line 645. How about the results reported in the publications? If the NH3 reduction treatments could enhance crop yield in previous studies?
Author Response
Line 28-29. The reductions presented in this sentences were achieved by trail hose application with and without slurry acidification, separately? The treatments, including NH3 reduction measures and crop seasons were not clearly explained in the abstract.
Reply: Thank you very much for this comment, the abstract has been reformulated to make these aspects of the study more clear.
Line 87-89. But if any negative impacts could be induced by acidification for slurry and soils?
Reply:This question was not the research topic of this study. But to give a full picture, negative aspects are covered in the discussion l 617 ff.
Line 96 and 101. Did you mean that previous field experiments measuring NH3 emissions including limited replicates? How many replicated plots are required for each treatment?
Reply: Some more information is added in lines 105 ff
Line 212-218. If these are commonly used by local famers in your study regions?
Reply: Yes, all slurry types and application methods are, with varying intensity, commonly used by local farmers. However, due to spreading of Corona desease by local on-farm Mink populations, most of the Minks were slaughtered in 2021 in Denmark. It remains an open question whether this industry will recover.
Lin 256. How about the measurement frequencies? How many days were needed before NH3 emissions from the slurry-added treatments decreased to the background levels?
Reply: Information is given in paragraph ll 260-274
Line 326. The method of trail hose application is the conventional treatment?
Reply: We do not find the reference to ‘conventional treatement’, in general the respective unacidified application technique was the reference treatment, in most cases unacidified trail hose application.
Line 370. Why did you set pH=6 as the target?
Reply: A pH reduction to 6.5-6 is usually recommended in practical systems as stated in ll 75-77
Line 384. What were the possible reasons?
Reply: L 384 is a figure heading, so we cannot reply to this comment.
Results shown in Tables 4 and 5 are complicated and difficult for the readers to understand.
Reply: We agree that the tables are complicated but they give the best summary on main effects and interactions of studied factors. To make the presentation of results simpler, we deleted the first model columns in Tab. 4 and 5 on tests of treatment effects, which we now only present for grain yield and ANRE in Fig. 6-9. The most relevant information emerges from the analysis of subdatasets on the effect of acidification, fertililiser type and application technologies and their interactions which are retained in the tables. Underlying data are included as supplementary material.
Line 456. In this section, I think the most important aspects are to discuss the underlying reasons for the reductions of NH3 emissions by application methods and acidification, and also the differences among slurry types. For example, how did the application methods affect NH3 emissions via changing soil NH4+ supply, and how about the soil pH in the treatments with vs. without acidification.
Reply: line 456 is a figure heading and this is still the results section. Discussion of effects is presented starting with l 485. There you find discussion of the importance of slurry properties on effectiveness of acidification.
In line 615-618, you indicated that PS exhibited low NH3 emissions and then acidification had no significant impacts. But the readers would more like to learn why PS emitted lower NH3. The effects of slurry type were one of the main research objective of this study. Suggest strengthening discussions on the differences of the internal properties of the slurry.
Reply: this is discussed in some detail in l 566 ff , due to the request to shorten the paper we do not want to elaborate more on this topic.
Line 628. The pH was measured before application, if the target pH had not achieved, why not add more acid?.
Reply: This was a test of the application system and the results show its limitations
Line 645. How about the results reported in the publications? If the NH3 reduction treatments could enhance crop yield in previous studies?
Reply: The main findings of the reports are summarized here and there is no more peer review published data on in-situ yield effects of acidification available.
Reviewer 2 Report
There is a lack of new information compared to the already known literature.
The results shown confirm what is already known in the literature. What is new in this work?
Author Response
Comments and Suggestions for Authors
There is a lack of new information compared to the already known literature.
The results shown confirm what is already known in the literature. What is new in this work?
Reply We generally agree to the criticism that the novelty of the research was not explained clear enough. The whole title, abstract and introduction was thoroughly revised to highlight the new contribution to research by this study.
The main novelties of this study consist in:
- Almost no data on in situ acidification on yields are available
- No published data on the efficacy of commercial acidification systems are available
- Usually systems are compared with respect on ammonia emission in time (i.e. by subsequent trials) and not simultaneously as in our study.
- No published data available on interaction of acidification with other application techniques
Round 2
Reviewer 2 Report
Authors improved their manuscript. I suggest authors to read this interesting paper: Maglione et al (2021). Aerated Buffalo Slurry Improves Spinach Plant Growth and Mitigates CO2 and N2O Emissions from Soil. Agriculture, 11(8), 758.
I warmly suggest to improve the "Introduction" section with other potential solutions reported in literature to mitigate soil NH3 emssion.
Author Response
Dear reviewer,
many thanks for your additional comment.
We have reworked an re-organized the introduction. Mitigation measures including slurry aearation as suggested by the reviewer are now addressed in ll. 63-78, new references (12,13) have been added and the reference list rearranged. We have also done few improvements to text and tables.
Sincerely yours
Andreas Pacholski